# Photosynthesis, Anatomy, and Metabolism as a Tool for Assessing Physiological Modulation in Five Native Species of the Brazilian Atlantic Forest

**DOI:** 10.3390/plants13141906

**Published:** 2024-07-10

**Authors:** Luis Alfonso Rodríguez-Páez, Mahmoud F. Seleiman, Bushra A. Alhammad, Yirlis Yadeth Pineda-Rodríguez, Marcelo F. Pompelli, Auxiliadora Oliveira Martins, Jaqueline Dias-Pereira, Wagner L. Araújo

**Affiliations:** 1Facultad de Ciencias Agricolas, Universidad de Córdoba, Montería, Córdoba 360002, Colombia; larguez@fca.edu.co (L.A.R.-P.); yadeth@fca.edu.co (Y.Y.P.-R.); 2Department of Plant Production, College of Food and Agriculture Sciences, King Saud University, P.O. Box 2460, Riyadh 11451, Saudi Arabia; mseleiman@ksu.edu.sa; 3Department of Crop Sciences, Faculty of Agriculture, Menoufia University, Shibin El-Kom 32514, Egypt; 4Biology Department, College of Science and Humanity Studies, Prince Sattam Bin Abdulaziz University, Al Kharj P.O. Box 292, Riyadh 11942, Saudi Arabia; b.alhamadi@psau.edu.sa; 5Departamento de Biologia Vegetal, Universidade Federal de Viçosa, Viçosa 36570-000, MG, Brazil; auxiliamartins82@gmail.com; 6Instituto de Ciências Biológicas e da Saúde, Universidade Federal de Viçosa, Rio Paranaíba 38810-000, MG, Brazil; jaqueline.dias@ufv.br

**Keywords:** amino acid metabolism, gas exchange, plant anatomy, metabolic pathway, phenotypic plasticity, upregulated of tricarboxylic acid metabolism (TCA)

## Abstract

The Brazilian Atlantic Forest, renowned for its exceptional species richness and high endemism, acts as a vital reservoir of terrestrial biodiversity, often referred to as a biodiversity hotspot. Consequently, there is an urgent need to restore this forest to safeguard certain species and to unravel the ecophysiological adaptations of others. This study aims to integrate some physiological parameters, including gas exchange and chlorophyll a fluorescence, with anatomical and metabolic techniques to elucidate how five different native species (*Paubrasilia echinata*, *Chorisia glaziovii*, *Clusia nemorosa*, *Licania tomentosa*, and *Schinus terebinthifolius*), each occupying distinct ecological niches, respond to seasonal variations in rainfall and their consequences. Our investigation has revealed that *C. nemorosa* and *P. echinata* exhibit robust mechanisms to mitigate the adverse effects of drought. In contrast, others demonstrate greater adaptability (e.g., *S. terebinthifolia* and *C. glaziovii*). In this context, exploring metabolic pathways has proven invaluable in comprehending the physiological strategies and their significance in species acclimatization. This study provides a comprehensive overview of the impact of water restrictions and their consequential effects on various species, defining the strategies each species uses to mitigate water privation during the dry season.

## 1. Introduction

The Brazilian Atlantic Forest encompasses a diverse mosaic of ecosystems, including mangroves and rainforests. This biome’s structure and floristic composition are notably distinct and influenced by several factors, including soil diversity, topography, and climates [1]. Over the past three decades, the Brazilian Atlantic Forest has become globally recognized as a top priority for biodiversity conservation. This unique and diverse biome, spanning across Brazil, Paraguay, and Argentina, has captured the attention of scientists, conservationists, and policymakers alike. With its exceptional range of species found nowhere else on Earth, the Atlantic Forest has rightfully earned its status as one of the world’s most crucial ecological hotspots [2]. Efforts to protect and restore this imperiled ecosystem have intensified, recognizing the urgent need to safeguard its rich biodiversity and preserve its invaluable ecosystem services for future generations. Unfortunately, Brazilian megalopolises’ devastation and urban development have resulted in the loss of more than 1,105,896 hectares of this precious biome [3], consisting of mature fragments exceeding three hectares with a closed canopy or showing no degradation detectable by satellite image. Regrettably, most of these remnants are small, exhibit high fragmentation, possess limited connectivity, and are characterized by secondary vegetation or non-forest structures. Additionally, the most critical fragments face the threat of further forest degradation, and their distribution is uneven across the biome’s different phytophysiognomies [4,5]. The escalating deforestation of the Brazilian Atlantic Forest has resulted in a significant loss of biodiversity. On a more positive note, collaborative efforts between the scientific community and ecologically motivated stakeholders have generated restoration initiatives that promise to mitigate the adverse effects of this degradation.

The exposure of plants to many environmental factors determines morphological, anatomical, and physiological variations contributing to species’ survival in different environments [6]. In that sense, the leaf is the most responsive organ to factors such as water availability [7]. Plant physiology and plant anatomy are two intertwined branches of botany that work together to unravel the intricacies of how plants function and adapt to their surroundings. Plant physiology delves into the fundamental processes within plants, such as photosynthesis, respiration, and transpiration, which are vital for their growth and survival. On the other hand, plant anatomy provides a detailed understanding of the internal structure of plants, including the arrangement of cells, tissues, and organs. Both fields complement each other, enabling scientists to explore the remarkable plasticity of plants—their ability to adjust and respond to environmental changes. Whether it is altering their growth patterns to reach for sunlight or adjusting their root systems to cope with limited water, the plasticity of plants showcases their resilience and resourcefulness in the face of diverse challenges. By comprehending the dynamic relationship between plant physiology, plant anatomy, and plasticity, researchers can gain profound insights into the fascinating world of plant life and contribute to various fields, from agriculture to environmental conservation. Some techniques for studying a leaf and its surrounding interaction media coexist. Still, the differential of this study is the physiological, anatomical, and metabolic study of native species never studied as proposed in this physiological focus as described in this study.

Metabolomics has additionally found great utility in the characterization of the metabolic response of plant cells to a range of biotic and abiotic stresses [8]. Primary messengers like glucose, fructose, and sucrose have long been recognized as critical molecular flags; raffinose has recently emerged as a significant player in unraveling the mechanisms of water stress resistance [9]. Its involvement in physiological adjustments opens up new avenues of exploration, shedding light on the intricate pathways through which organisms adapt and thrive in challenging environments [10,11]. Bazzaz and Carlson [12] proposed that species that thrive under the tree canopy have a metabolism that operates constantly to keep the organism continually adapted to low levels of light and resources. On the other hand, these authors suggest that organisms that depend on clearings have greater metabolic flexibility and, therefore, a greater capacity to respond to environmental variations.

This study evaluated the photosynthetic, anatomical, and metabolic plasticity in response to different rainy seasons of five tropical tree species belonging to different successional groups. All of these species are commonly used in recent forest restoration programs in Brazil. We hypothesize that the plasticity capacities to cope with environmental alterations must occur after the remodeling of the gas exchanges, tissue allocation, and metabolic changes. Our main goal is to develop a logical explanation in detail of the morphoanatomical and metabolic cues able to permit the plant abilities to cope with different seasons, here referred to the rainy season and dry season in a Brazilian Atlantic Forest fragment in Paulista City, Pernambuco, Brazil.

## 2. Results

### 2.1. Climatological Data

The precipitation in the rainy season was consistently high (212 ± 7 mm per month, totaling 847 mm; Figure 1), while the precipitation was relatively low in the dry season (76 ± 2.3, totaling 607 mm) with a decrease in precipitation on average by about 28% from the wet season to the dry season. The air humidity remained relatively constant between seasons (74 ± 3%) during the experimental year. The mean global radiation (daily integrated over the day) fluctuated according to the respective season, with lower mean global radiation between March and August (early and late rainy season, 17 MJ m^−2^) and between August and February (23 MJ m^−2^). The mean temperature was almost stable over the seasons, being 24.8 ± 0.5ºC during the rainy season and 24.7 ± 0.8 during the dry season (Figure 1).

### 2.2. Studied Species

All studied species are listed in Table 1, where some characteristics of each species are described as photosynthetic metabolism type, successional group, economic importance, and tree information. *Chorisia glaziovii* and *Clusia nemorosa* are pioneers, while *Licania tomentosa* is late successional. *Paubrasilia echinata* and *Schinus terebinthifolia* are described as pioneer or early successional (*S. terebinthifolia*) or late successional (*P. echinata*). All species are classified as having photosynthetic metabolism C3 type; however, only one paper describes *C. nemorosa*, and thus, there is a need for further confirmation since the vast majority of *Clusia* species are classified as C3-CAM or CAM species (Table 1).

### 2.3. Gas Exchange and Chlorophyll a Fluorescence

Before analyzing gas exchange, we determined the specific leaf area (SLA; Table 2 and Table 3). As described in Table 2 and Table 3, there are differences in the SLA of all analyzed species (*p* < 0.001); however, this difference was not attributed to the season of sample months (*p* = 0.435). After that, we verified that the SLA ranged from 5.1 cm^2^ g^−1^ (*C. nemorosa*) to 15.1 cm^2^ g^−1^ (*C. glaziovii*). Thus, all gas exchange data were expressed on a fresh weight basis instead of an area-based one since *C. nemorosa* has more photosynthetic tissue per leaf area than *C. glaziovii*. If the gas exchange results were evaluated on an area basis, the net photosynthesis (*P*_N_) data would be underestimated as the SLA decreases or increases the leaf thickness, leading to the misinterpretation of the gas exchange data.

Figure 2A shows that the *P*_N_ starts high in all species and decreases during the dry season. In *P. echinata*, *P*_N_ was 71.6 μmol CO_2_ g^−1^ s^−1^ in June but fell sharply to 29.8 μmol CO_2_ g^−1^ s^−1^ in February, a decrease of 58.4%. A similar pattern was described for *S. terebinthifolia* since its *P*_N_ fell from 118 μmol CO_2_ g^−1^ s^−1^ in June to 48 μmol CO_2_ g^−1^ s^−1^ in February, a decrease of 59.3%. In *C. glaziovii*, the decline was more abrupt since the *P*_N_ fell from 153.7 μmol CO_2_ g^−1^ s^−1^ in June to 51.2 μmol CO_2_ g^−1^ s^−1^ in February (Figure 2), a fall of ~67%. In *L. tomentosa* the *P*_N_ fell from 125.8 μmol CO_2_ g^−1^ s^−1^ in June to 56.7 μmol CO_2_ g^−1^ s^−1^ in December, and after an increase to 91 μmol CO_2_ g^−1^ s^−1^ in February. On the other hand, *C. nemorosa* shows a *P*_N_ that is relatively stable but smaller than that of other species. A similar tendency was shown for stomatal conductance (*g*_s_) since that is of a higher magnitude. While *C. glaziovii* shows an abrupt decrease in *g*_s_ of 78%, *P. echinata* and *S. terebinthifolia* show a less abrupt reduction in their *g*_s_, a reduction of 58% and 30%, respectively. The *g*_s_ values of *L. tomentosa* show a similar tendency to that described for *P*_N_, while *C. nemorosa* shows a decrease in *g*_s_ to 69.3% (Figure 2B).

The integrity of the photosystems within the evaluated period can be verified using chlorophyll a (Chl a) fluorescence parameters (Figure 3A). Thus, it appears that the species *C. nemorosa*, *L. tomentosa*, and *S. terebinthifolia* presented an Fv/Fm within the expected values for healthy plants (~0.78). On the other hand, the species *C. glaziovii* showed an Fv/Fm= of 0.53 in December, a reduction of 26% from the reference value in June. The reduction in Fv/Fm is considered a good response in our analysis because it can be used to confirm the hypothesis of photoinhibition in *C. glaziovii*, which shows decreases in fluorescence parameters like Fv/Fm. The species *P. echinata* also showed a decrease in the modulation of the Fv/Fm in the driest months since the value measured in December was 0.64, 13.5% lower than the value shown in June (0.74).

The dissipation of energy from photons captured by the antenna complexes in the form of heat (Figure 3B–D) is another indication of an overloaded photosystem. In this sense, we found that all species showed greater heat dissipation in the driest months of the year, with emphasis on the *S. terebinthifolia* species, which increased its D from 0.37 (in June) to 0.82 (in February), an increase of 2.2-fold if we consider the June value; however, considering the October values, at the beginning of the critical dry period, this increase is 4.1-fold. *C. nemorosa* was another species that greatly increased the release of energy in the form of heat as it fluctuated from 0.43 (in June) to 0.94 in December and to 0.65 in February, a significant increase of 2.2- and 1.5-fold, respectively. *P. echinata*, *C. glaviovii*, and *L. tomentosa* showed a decrease in D in October. Still, the February value is not significantly different from the initial value in June, demonstrating that the stress due to the water deficit may have been lower in these species. It is known that these species have a more robust root system, which may have influenced the plant’s resistance in the most critical periods of the year after an abrupt decrease in precipitation was shown in October. The current PSII efficiency (ΦPSII; Figure 3C) during exposure to sunlight showed that, except for the species *L. tomentosa* and *S. terebinthifolia*, the other species did not show a significant change in ΦPSII. The ΦPSII values obtained in February for the species *L. tomentosa* and *S. terebinthifolia* are 39.3% and 63.2% lower than those verified in August. The electron transport rate (ETR; Figure 3D) showed a different curve than expected since the highest values were measured in the driest months, and the values obtained in February are not significantly different from the values captured in June, except the species *P. echinata* which showed a 51% lower ETR in February compared to June.

### 2.4. Photosynthetic Pigments

To a greater or lesser extent, all species showed a decrease in chlorophyll a, b, and total in the driest months. Chlorophyll a (Figure 4A) was drastically reduced in February compared to June, being 92.8% in *C. glaziovii* and 82.3% in *P. echinata*. In *S. terebinthifolia* (41.4%) and *C. nemorosa* (26.6%), the reduction in February was moderate, while in *L. tomentosa*, the reduction was weak (4.8%). In the same period, the reduction in Chl b was similar to that presented for Chl a, where the species *C. glaziovii* and *P. echinata* showed a very expressive reduction in chlorophyll b in the order of 91.5% and 78.6%, respectively, considering July and February only. *S. terebinthifolia* showed a moderate reduction in Chl b (52%), while in *C. nemorosa*, the reduction was weak but significant (11.3%). On the other hand, in *L. tomentosa*, the Chl b concentration increased (1.12-fold) in February compared to June. By analogy, the total chlorophyll concentration showed the same pattern as the other measurements, with *C. glaziovii* and *P. echinata* showing the most significant reductions (92.5% and 80.6%, respectively) in the driest months (February). *S. terebinthifolia* showed a less accentuated reduction (48.7%), while in *C. nemorosa* and *L. tomentosa*, the reduction was 19.8% and 1.5%, with the latter value not being significantly different from the concentration measured in June. The concentration of total carotenoids measured in February was 47.9% less than that in June in *P. echinata*, while in *S. terebinthifolia*, the reduction in February was not significant (2.4%). In the species *C. nemorosa*, *L. tomentosa*, and *C. glaziovii*, the concentration of total carotenoids measured in February was 1.37-, 1.28-, and 1.24-fold higher when compared to the values described in June. Also, the total chlorophyll (Chl a + b) to carotenoids (Car) ratio decreased in February compared to June. The reduction was from 1.06 to 0.39, 4.46 to 0.27, 0.86 to 0.50, 0.86 to 0.66, 0.86 to 0.45 in *P. echinata*, *C. glaziovii*, *C. nemorosa*, *L. tomentosa*, and *S. terebinthifolia*, respectively.

### 2.5. Plant Anatomy

Anatomically, *P. echinata* has a thin external periclinal wall, unistratified in adaxial (ADA) and abaxial (ABA) epidermis surfaces. In the ADA surfaces, the cells are more tabular when compared to the cells in the ABA surfaces (Figure 5A,B). The mesophyll is a dorsiventral, collateral bundle, and it is hypostomatic with one layer of palisade parenchyma (PP) and six to eight layers of spongy parenchyma (SP). *C. glaziovii* showed a thin external periclinal wall, unistratified epidermis, two layers of PP, four layers of SP with vast intercellular spaces, a hypostomatic leaf, and secretory channels (Figure 5C,D). In the dry season, the leaves also had a thicker adaxial periclinal wall, unistratified epidermis, two layers of PP, eight layers of SP, and secretory channels, in addition to more significant thickening of the leaf. In *C. nemorosa*, a very well-developed adaxial hypodermis can be observed in all images (Figure 5E,F), while an abaxial surface was described only on dry plants. The epidermis of ADA and ABA is unistratified with tabular cells, two subepidermal layers facing the ADA, dorsiventral mesophyll, collateral bundle, and hypostomatic. In the dry season, the external periclinal wall is thicker, both in the ADA and the ABA. The subepidermis of ADA and ABA is unistratified, with tabular cells, 2 subepidermal layers facing the ADA and 1 subepidermal layer facing the ABA, dorsiventral mesophyll, and approximately 16 layers with collateral bundles, hypostomatic leaves, and a thicker mesophyll (Figure 5). *L. tomentosa* had a thin external periclinal wall and unistratified ADA and ABA (Figure 5G,H). The ADA cells are square, while in ABA, the cells are tabular. The mesophyll is an isobilateral, collateral bundle, and it is hypostomatic. A thick external periclinal wall was also evidenced in the samples collected in the dry season. The ABA cells were slightly less tabular than those described in the leaves of the rainy season. *Schinus terebinthifolia* has a thin external periclinal wall, a uniseriate epidermis on ADA and ABA, a subepidermal layer facing only the epidermis on the adaxial surface, a dorsiventral mesophyll with two layers of PP, approximately four layers of SP and two layers facing the ABA tending to PP, collateral bundle, hypostomatic leaf (Figure 5I,J). The leaves collected in the dry season showed a thick external periclinal wall, unistratified ADA and ABA, a subepidermal layer, a mesophyll with two layers of PP, and approximately six layers of SP.

Micromorphometrically, the ADA (Figure 6A) was thinner in *C. nemorosa*, intermediate in *C. glaziovii* and *S. terebinthifolia*, and thicker in *P. echinata* and *L. tomentosa*. It should be noted that the occurrence of a thick adaxial subepidermis in *C. nemorosa* and *S. terebinthifolia* (Figure 6E) is a fact that may have contributed to the reduction in ADA in these two species. Moreover, these values differ for leaves collected in the rainy season and those collected in the dry season. In leaves collected in the rainy season, the PP (Figure 6B) had a median thickness ranging from 29.3 μm in *C. nemorosa* to 39.8 in *P. echinata* or greater thickness in *L. tomentosa* (85.5 μm) and in *S. terebinthifolia* (95.4 μm). However, this profile showed a slight alteration in the leaves collected in the dry season, where the PP thickness ranged from 35.5 μm (*C. glaziovii*) to 92.6 μm (*S. terebinthifolia*). The SP (Figure 6C) was highly reduced in *C. glaziovii* (28.2 μm) to extremely thick (318.5 μm) in *C. nemorosa*; these values did not show a change in the trend in the dry season, where the SP thickness in the plants collected in the dry season was 6.5% (*P. echinata*), 17.7% (*C. nemorosa*), 30.3% (*L. tomentosa*), and 34.6% (*C. glaziovii*) thicker than those measured in the rainy season. The SP of *S. terebinthifolia* was, unlike the others, 26.3% finer in the leaves collected in the dry season when compared to those collected in the rainy season (Figure 6C). In this way, the SP showed a much greater thickness than the PP in increasing order among the rainy-season plants: *L. tomentosa* (1.05-fold), *S. terebinthifolia* (1.56-fold), *P. echinata* (2.67-fold), and *C. nemorosa* (10.88-fold). In *C. glaziovii*, the SP was 13.9% thinner than the PP. In leaves collected in the dry season, the SP was 1.19-fold (*S. terebinthifolia*), 1.22-fold (*C. glaziovii*), 1.67-fold (*P. echinata*), 1.83-fold (*L. tomentosa*), and 10.32-fold (*C. nemorosa*) thicker than PP. Moreover, *C. nemorosa* showed the highest thickness of SP, 1.2-, 1.6-, 1.7-, and 3.2-fold higher than *S. terebinthifolia*, *L. tomentosa*, *P. echinata*, and *C. glaziovii* (Figure 4B). The ABA (Figure 6D) thickness was distinct in the rainy and dry seasons. In the rainy season, the ABA ranged from 8.8 μm (*C. nemorosa*) to 19.2 μm (*L. tomentosa*), while in the dry season, the ABA ranged from 9.3 μm to 18.5 μm in the same species. Also, *C. nemorosa* presented a thicker ADA (Figure 6F). Considering all tissues, the rainy leaves have the following order of total leaf thickness (TLT): *C. glaziovii* (82.1 μm), *P. echinata* (181.1 μm), *L. tomentosa* (231.3 μm), *S. terebinthifolia* (266.8 μm), and *C. nemorosa* (436.7 μm). In the dry season, the TLT was 106.7 μm (*C. glaziovii*), 222.1 μm (*P. echinata*), 242.5 μm (*S. terebinthifolia*), 257.9 μm (*L. tomentosa*), and 555.7 μm (*C. nemorosa*) (Figure 6G).

Morphologically, all evaluated species are hypostomatic. In *P. echinata* and *C. nemorosa*, the stomata are paracytic with kidney-shaped guard cells. In *C. glaziovii*, the stomata are paracytic with abundant deposition of waxes, covering part or almost all of the ostiole. In *L. tomentosa*, the stomata are staurocytic with stomata located slightly above the other epidermal cells, and in *S. terebinthifolia* the stomata are paracytic, showing very large opened ostioles in rainy season and abundant waxes in the dry season. Also, in the dry season, these stomata often are covered with dense absorbent trichomes. The stomatal density ranged from 62.4 mm^−2^ (*L. tomentosa*) to 173.2 mm^−2^ (*S. terebinthifolia*) in rainy-season plants and from 74.1 mm^−2^ (*L. tomentosa*) to 165.9 mm^−2^ (*C. nemorosa*) in dry-season plants (Figure 7A). In ABA, the ordinary cell density ranged from 666.2 mm^−2^ (*C. glaziovii*) to 1343.9 mm^−2^ (*S. terebinthifolia*) in the rainy season and from 768.2 mm^−2^ (*L. tomentosa*) to 1299.9 mm^−2^ (*S. terebinthifolia*) in dry-season plants (Figure 7B).

The regressions between stomatal density (SD) and ordinary cell density (OCD) (Figure 8A) or between stomatal density and stomatal index (SI) (Figure 8B) are presented. Except for the regression between SD and OCD for *C. glaziovii*, all other regressions were significant, with a high regression coefficient and high *p* value. As there was a high correlation between SD and OCD and an even higher regression between SD and SI, the SI modulation could be due to both higher SD and SI, but as the variation of SD (5-fold) was greater than that of OCD (4.2-fold), SD is believed to be modulating SI rather than OCD.

### 2.6. Plant Metabolism

In response to the water deficit caused by seasons, considerable changes in the levels of a wide range of organic acids, amino acids, and sugars were evident (Figure 9). To provide an overview of the major metabolic changes observed, putative metabolic pathways were drawn (Appendix A). Forty-eight metabolites were detected with GC-MS. Twelve organic acids, eight sugars, twenty-one amino acids, four sugar alcohols, and three secondary metabolic compounds were identified and compared between plant species (Figure 9). To help us understand the metabolism and metabolic pathway, the species were presented separately and after in a correlation in an integrated form.

#### 2.6.1. *Paubrasilia echinata*

In *P. echinata* plants measured in February, some compounds were downregulated, such as the organic acids citrate, dehydroascorbate, gluconate, glycerate, malate, succinate, and the sugars fructose, glucose-6-phosphate, and sucrose, as well as the amino acids asparagine, aspartate, β-alanine, glutamine, histidine, homoserine. Alternatively, cysteine, methionine, and phenylalanine were upregulated. Phenylalanine was detected at high relative abundance (more than 1.5-control-fold change [FC]). In contrast, citrate, dehydroascorbate, gluconate, glycerate, malate, malonate, succinate, fructose, sucrose, and tyrosine were detected at low relative abundance (less than 0.5 FC). Significantly, raffinose was upregulated, as were the other sugars with osmoprotective functions. These observations do not lead us to a defined metabolic path, but a remobilization of some sugars in favor of key amino acids with a low capacity to build osmoregulators (Appendix A) is indicative of the TCA metabolism being reinforced in favor of amino acid production through carbon skeletons. So, the water deficit was severe in this species and may have caused a decrease in *P*_N_.

#### 2.6.2. *Chorisia glaziovii*

In *C. glaziovii* water-stressed plants, the organic acids 3-P-glycerate, citrate, dehydroascorbate, malate, malonate, succinate, fructose, fructose-6-P, maltose, mannose, and sucrose and the amino acids asparagine, aspartate, glutamine, glycine, histidine, homoserine, hydroxyproline, tryptophan, and tyrosine were downregulated. In addition, α-ketoglutarate, fumarate, glucose, β-alanine, isoleucine, leucine, and methionine were upregulated. These findings lead us to a putative metabolic pathway of remobilizing some sugars and amino acids in favor of structural amino acids and those involved in secondary metabolisms, such as β-alanine with a deviation of α-ketoglutarate (Appendix A).

#### 2.6.3. *Clusia nemorosa*

In *C. nemorosa*, the metabolic profile was strongly upregulated and presented a pattern completely different from that observed in the other species evaluated here. In general, most compounds, especially amino acids, organic acids, and sugar alcohols increased following the beginning of the evolution of water stress. This pattern, together with the strong increase in myo-inositol, erythritol, raffinose, trehalose, and shikimate lead us to believe that structural amino acids, as well as compounds from the glycolytic pathway and of the tricarboxylic acid metabolism (TCA), are being diverted towards the synthesis of defense compounds (secondary metabolism), osmoregulation, and synthesis of aromatic amino acids (Appendix A). Patterns like this can be indicative of a metabolic reprogramming capable of keeping the plant physiology stable, as can be observed in gas exchanges (Figure 2A,B).

#### 2.6.4. *Licania tomentosa*

In *L. tomentosa*, the metabolic profile presented expressive changes in water stress. A strong reduction was observed in 3-P-glycerate, citrate, dehydroascorbate, gluconate, malate, fructose-6-P, glucose, histidine, and ornithine. At the same time, a significant increase of more than 1.5-FC was measured for glycerate, succinate, maltose, sucrose, glycine, phenylalanine, and erythritol. These modifications can provide evidence of an activation of the metabolism of aromatic amino acids and osmoregulatory compounds (Appendix A).

#### 2.6.5. *Schinus terebinthifolia*

*S. terebinthifolia* presented a strong reduction in gluconate, glycerate, succinate, GABA, aspartate, β-alanine, citrulline, glycine, homoserine, hydroxyproline, isoleucine, leucine, methionine, phenylalanine, trehalose, and spermidine. In the same way, a strong increase was observed in α-ketoglutarate, isocitrate, glucose, histidine, ornithine, serine, and tryptophan. This metabolic pattern shows a species that probably remobilizes some carbon compounds for synthesizing aromatic amino acids and other structural amino acids, as well as those involved in urea excretion (Appendix A). This alternative route is characteristic of a species that has not invested in costly carbon skeletons to acclimate it to environmental conditions, a fact shown by the strong reduction in *P*_N_, exacerbated increase in heat (D), and decrease in Φ_PSII_.

In *C. glaziovii* (Appendix A) and *C. nemorosa* (Appendix A), higher levels of the tricarboxylic acid (TCA) cycle and glycolytic pathway intermediates such as glucose, pyruvate, α-ketoglutarate, succinate, and fumarate were observed in plants under water stress. The changes in the malate content of mature leaves from each of the species during the rainy season and the dry season (Figure 9) reflected the gas exchange characteristics illustrated in Figure 2. In this sense, *P. echinata*, *C. glaziovii*, *C. nemorosa*, and *L. tomentosa* depleted their malate concentration as the water deficit increased in accordance with the seasonal rainfall gradient. In another way, *S. terebinthifolia* plants significantly increase their malate concentration. Thus, the water deficit, demonstrated in February, promoted significant alterations in levels of malate in mature leaves of all studied species reported here, where plants of *P. echinata*, *C. glaziovii*, *C. nemorosa*, and *L. tomentosa* depleted their malate concentration from 53% to 65%, from 56% to 61%, from 3 to 51%, and from 28% to 48%, respectively. Alternatively, *S. terebintifolia* promotes malate accumulation 1.2- to 1.6-fold more in comparison to the control plants evaluated in the middle of the rainy season. Water stress leads to a decrease in *P*_N_ (in all plant species, except *C. nemorosa*). The substantial decrease in *P*_N_ was accompanied by a substantial decrease in the overall content of citrate compared to that in well-watered plants (Figure 9; Appendix A). In general, the levels of the TCA cycle intermediates exhibited relatively high changes during water stress, but these changes were species-specific. Also, in three of the five species (*P. echinata*, *C. glaziovii*, and *S. terebinthifolia*), the concentration of trehalose, a compound involved with osmotic adjustment, decreased with drought. In contrast, in *C. nemorosa*, the level of trehalose was strongly increased. Raffinose and trehalose strongly increased in *C. nemorosa* plants, with an increase in water stress evolution. In all other species, both sugar alcohols were weakly (*S. terebintofolia*) or strongly (*P. echinata* and *C. glaziovii*) decreased. In *L. tomentosa*, these sugar alcohols did not show any significative (*p* > 0.05) change along water stress. Fructose, as well, was another sugar that strongly increased in *C. nemorosa* due to the water stress. Glucose (*C. glaziovii* and *S. terebinthifolia*) and glucose-6-P (*S. terebinthifolia*) were two sugars with a substantial increase in response to water stress promoted by lesser rainfall.

Only in *C. nemorosa* was the shikimate pathway most strongly increased. However, in *P. echinata*, *C. glaviovii*, and *S. terebinthifolius*, the pathway was moderately decreased (Figure 9; Appendix A). Histidine, an essential amino acid that becomes incorporated into proteins and is required for cell growth and reproduction, was moderately depleted in *P. echinata* and *L. tomentosa* and strongly depleted in *C. glaziovii*, while in *C. nemorosa* and *S. terebinthifolia*, it was moderately and strongly increased, respectively.

After a more individual analysis, we investigated the interaction between the compounds using Pearson’s correlation. Myo-inositol shows a positive correlation with raffinose (*r* = 0.805) and a negative correlation with trehalose (*r* = −0.473), while trehalose and raffinose both show a negative correlation (*r* = −0.342) between them (Appendix A). A positive correlation of *P*_N_ and serine (*r* = 0.454) links *P*_N_ with the amino acid pool, reinforced by a negative correlation between *P*_N_ and tryptophan (*r* = −0.316) or tyrosine (*r* = −0.527). In another way, the negative correlation between *P*_N_ and erythritol (*r* = −0.460) and myo-inositol (*r* = −0.526) leads us to argue that a decrease in *P*_N_ in response to a water deficit activates the osmoregulation mechanism (Appendix A). The similar correlation using *g*_s_ for the same metabolic pathways reinforces this argument. The chlorophyll a fluorescence correlations confirm that *P*_N_ is related to metabolism, reinforced by a positive correlation between rainfall and *P*_N_ (*r* = 0.350) and *g*_s_ (*r* = 0.366). The positive correlations between maltose and Φ_PSII_ (*r* = 0.324) and ETR (*r* = 0.249) and between mannose and ETR (*r* = 0.230) reinforce a deviation of the reduction power to increase osmoregulators, like mannose and maltose.

The negative correlation between 2-oxoglutarate and malate (*r* = −0.873), succinate (*r* = −0.825), fructose (*r* = −0.790), glucose-6-phosphate (G-6-P) (*r* = −0.881), and maltose (*r* = −0.388) or between 3-phosphoglycerate and malate (*r* = −0.661), succinate (*r* = −0.627), fructose (*r* = −0.624), and G-6-P (*r* = −0.664) and the positive correlation between 3-phosphoglycerate and fumarate (*r* = 0.392) and glucose (*r* = 0.242) denotes that the TCA was downregulated, perhaps as a part of a function to increase the secondary metabolism due to high positive correlation between 3-phospho glycerate and GABA (*r* = 0.646) (Appendix A). The negative correlation between citrate and 2-oxoglutarate (*r* = −0.848) or a positive correlation between citrate and malate (*r* = 0.941), succinate (*r* = 0.882), fructose (*r* = 0.916), and G-6-P (*r* = 0.922) confirms the hypothesis that the TCA is downregulated, maybe in the function of osmoprotection with maltose (*r* = 0.341) and mannose (*r* = 0.276) to the detriment of a secondary metabolism with GABA (*r* = −0.792). Also, fumarate presented a negative correlation with malate (*r* = −0.728), succinate (*r* = −0.686), fructose (*r* = −0.649), and G-6-P (*r* = −0.733); malate showed a positive correlation with succinate (*r* = 0.882) and fructose (*r* = 0.929). The positive correlation between glycerate and malate (*r* = 0.887), succinate (*r* = 0.943), fructose (*r* = 0.847), and G-6-P (*r* = 0.914) indicates coordination between the glycolytic pathway and TCA. Furthermore, a strong positive correlation between palmitate and succinate (*r* = 0.885), fructose (*r* = 0.940), and glucose (*r* = 0.988) seems to be a link between beta-oxidation of fatty acids and an increase in TCA (Appendix A). The high negative correlation between malate and GABA (*r* = −0.815) or malate and asparagine (*r* = −0.287) is indicative of an interplay between TCA and secondary metabolism and amino acid metabolism. The positive correlation between succinate and maltose (*r* = 0.321) and mannose (*r* = 0.337) and a negative correlation between succinate and GABA (r = −0.767) or asparagine (r = −0.279) confirm this deviation of TCA metabolites with secondary metabolism and amino acid metabolism.

The strong negative correlation of 2-oxoglutarate with tryptophan (*r* = −0.650) and tyrosine (*r* = −0.887) denotes an apparent deviation of TCA metabolites for the synthesis of aromatic amino acids (Appendix A). Elsewhere, the positive correlation of 2-oxoglutarate with erythritol (*r* = 0.715), myo-inositol (*r* = 0.882), raffinose (*r* = 0.749), and trehalose (*r* = 0.410) leads to us to argue that 2-oxoglutarate deviated to synthesize osmoregulators following rainfall. This highlights the negative correlation between rainfall and 2-oxoglutarate (*r* = −0.428). A similar pattern was verified with 3-P-glycerate, which shows a negative correlation with tryptophan (*r* = −0.504) and tyrosine (*r* = −0.671) and a positive correlation with erythritol (*r* = 0.605), myo-inositol (*r* = 0.668), raffinose (*r* = 0.572), and trehalose (*r* = 0.555).

Fructose shows a higher negative correlation with asparagine (*r* = −0.246), cysteine (*r* = −0.387), glycine (*r* = −0.387), histidine (*r* = −0.212), homoserine (*r* = −0.237), isoleucine (*r* = −0.417), methionine (*r* =−0.652), and serine (*r* = −0.687), which supports our hypothesis of deviation or TCA intermediaries to amino acid synthesis. Moreover, the negative correlation between fructose and phenylalanine (*r* = −0.399), tryptophan (*r* = −0.864), and tyrosine (*r* = −0.927) denotes a deviation of fructose to aromatic amino acid synthesis. The positive correlation of fructose with erythritol (*r* = 0.800), myo-inositol (*r* = 0.942), raffinose (*r* = 0.753), and trehalose (*r* =0.452) leads us to speculate that fructose deviates to osmoregulator synthesis, a hypothesis supported by similar correlations between succinate and the same metabolites.

In all plant species, we verify a negative correlation between tryptophan and β-alanine (*r* = −0.401) and between glutamate and β-alanine (*r* = −0.437), perhaps indicating an interplay of some structural amino acids with secondary metabolism.

Furthermore, a small OCD was inversely correlated with serine (*r* = −0.769) and directly correlated with tryptophan (*r* = 0.704), tyrosine (*r* = 0.908), erythritol (*r* = 0.842), and myo-inositol (*r* = 0.904). These correlations lead us to argue that more compacted leaves (smaller) as a consequence of rainfall provoke a deviation to structural amino acids to the detriment of osmoregulators like erythritol and myo-inositol.

### 2.7. Principal Component Analysis

The first two principal components (PC1 + PC2) of the PCA explain 85.2% of the multivariate variability of this group of variables (Figure 10). Five metabolites are composed of the best traits to explain the variability associated with *P*_N_: mannose (52.6%), dehydroascorbate (49.7%), histidine (48.7%), asparagine (45.7%), and leucine (43.6%). After clustering, we describe the formation of five monophyletic groups, each containing one plant species, both in rainy and dry conditions (Figure 10). In another way, metabolites such as tyrosine (25.1%), palmitate (25%), serine (24.8%), citrate (23.9%), and raffinose (23.2%) are the features that contribute less to *A*_N_ modulation between rainfall gradients (Appendix A). The anatomical features like palisade parenchyma (45.6%) and SLA (35.6%) contribute highly to the modulation of *A*_N_ (Appendix A).

### 2.8. Plasticity Index

After individually analyzing all the plasticity indices (Table 4), the indices were grouped into physiological, anatomical, and metabolic features. The highest average plasticity indices (0.791) were described within the metabolite group. The lowest average plasticity indices were found in the anatomical features (0.456), and the physiological features had an intermediate index value (0.662). Within the physiological features, *P. echinata* (0.847) and *C. glaziovii* (0.828) were more significantly plastic, while *C. nemorosa* (0.548) and *L. tomentosa* (0.501) exhibited less plasticity. The *S. terebinthifolia* showed intermediate plasticity (0.854) and did not differ from either group. The *g*_s_ and *P*_N_ showed the highest plasticity index, which is highlighted.

Within the anatomical features, *S. terebinthifolia* (0.617) and *L. tomentosa* (0.497) were the most plastic, differing from *C. nemorosa* (0.372) and *P. echinata* (0.349). When we analyzed each variable individually, the lowest and the highest indices were found to be of ordinary cell density (0.251) and abaxial epidermis surface thickness (0.715), respectively.

The plasticity index involving the metabolic compounds showed greater amplitude values. The species *C. glaziovii* (0.837), *C. nemorosa* (0.833), and *S. terebinthifolia* (0.812) showed the highest plasticity indices and did not differ from each other, while *P. echinata* (0.723) and *L. tomentosa* (0.734) were the smallest (Table 4). Shikimic acid (0.957) and aspartate (0.959) were the highlights in all species, while palmitate (0.616) and serine (0.628) showed lower plasticity index values.

In our study, all sugar alcohols containing higher plasticity indices were from all groups of metabolites and included all species ranging from 0.504 to 1.0 and a mean value of 0.803. The lower plasticity index in *L. tomentosa* and *S. terebinthifolia* supports the hypothesis of restrictions in morphological plasticity for late secondary species [26,27], which states that shade-tolerant species would exhibit less morphological plasticity due to the longer leaf longevity influencing the speed of tracking environmental changes.

## 3. Discussion

In the present study, we provide a seasonal profile of the diurnal photosynthetic performance of *Paubrasilia echinata*, *Chorisia glaziovii*, *Clusia nemorosa*, *Licania tomentosa*, and *Schinus terebinthifolia* grown under water deficit conditions, which promoted significant alterations in morphophysiological and metabolic traits. The carbon assimilation by the leaves is influenced by physiological and morphological adjustments, in addition to changes in the characteristics related to photosynthesis, such as chlorophylls and total carotenoid concentrations. The leaf thickness results in an increase or decrease in the specific leaf area (SLA) [28]. In this study, the five species exhibited different SLAs, and the gas exchange data were represented by leaf mass, not by leaf area, since SLA indicates that in the same leaf area, there are different thicknesses and, thus, more or fewer photosynthetic tissues.

Tropical plants growing under water stress exhibit modulations in photosynthetic processes, including disruptions in the stomatal control of the gas exchange [29,30], changes in the kinetics of chlorophyll a fluorescence [31], damage to photosynthetic components [32], and alterations in carbohydrate status [33,34]. Additionally, precipitation effects can alter the performance of the photosynthetic apparatus throughout the day regardless of the seasonal rainfall regime. This fact was corroborated by different vegetative phytophysiognomies that can lead to similar net photosynthesis (*P*_N_) in *C. glaziovii*, *L. tomentosa*, and *S. terebinthifolia*. Also, we demonstrated that *C. nemorosa* does not change its *P*_N_ under extreme water deficit in December and February. This pattern was previously described *Spondias tuberosa* as a savanna-like species [35].

Chlorophyll and carotenoid concentrations can vary depending on the intensity of irradiance. Boeger et al. [36] describe that chlorophyll molecules could control photosynthetic rates by absorbing light energy, while carotenoids are capable of dissipating excess energy. Therefore, higher levels of chlorophylls are expected in leaves under lower light intensity, and higher levels of carotenoids in leaves exposed to full irradiance. In this study, we considered full-sun samples as those collected from plants in the dry season and low-irradiance samples as those collected in the rainy season, following the precipitation and irradiance profile shown in Figure 1. Mengarda et al. [11] describe that the Chl/Carot in *P. echinata* was decreased in shade leaves when transferred to full sun [11].

Plant acclimatization can be studied in several ways: (i) monitoring chlorophyll a fluorescence and measuring the photosynthetic rate; (ii) anatomical changes leading to increased assimilation rates that can also occur in fully developed leaves experiencing increase and decrease in light levels (iii) changes in metabolism in general, carbon metabolism (derived from molecules shifted to TCA) and nitrogen in particular (synthesis of amino acids, proteins, and non-protein amino acids) [34]. For example, the higher PP in the *S. terebinthifolia* would be related not only to of low water availability conditions but also to the higher light intensities [6]. Adaptations in response to different light intensities have also been reported by Dias et al. [37] and Pompelli et al. [38]. The greater thickness of the parenchyma, especially PP, can be considered a strategy to maximize the *P*_N_ even with the high intensity of light received, as in the case of *C. nemorosa*. In *C. nemorosa*, the PP presented with 2 to 3 layers of cells, distinct from what was described by Fernandes [39] as a single palisade layer. On the other hand, the greater thickness of the SP can be considered an escape strategy to very stressful environments, as a less denser SP can facilitate cooling of tissues, contributing to heat loss with less water loss to the atmosphere [40,41].

Fernandes [39] describes non-significative differences between leaf anatomy in the rainy and dry seasons in *C. criuva*, data that diverges from those presented in this study for *C. nemorosa*, where significant anatomical differences are described in the leaves of the rainy and dry seasons. *Clusia* minor [42] and other *Clusia* sp. have the ability to shift their characteristic CAM metabolism during the dry season to C3-type metabolisms in the wet season. In *Clusia* sp., the induction of CAM appears to be triggered by conditions that promote the breakdown of organic acids during the day [43]. C3 or CAM metabolism may vary within the same species depending on its cultural practices, that is, during drier or rainy periods, during the night, dawn/dusk [42,44]. There are many possibilities of expression of C3-CAM metabolism; therefore, specific studies in each species and in given cultivation conditions are interesting and need to be carried out.

An increase in the total leaf thickness (TLT), ADA, PP, SP and SD was previously reported in *P. echinata* acclimated to full sun (here compared to the dry season) in comparison to shade leaves [11]. Gama et al. [45] concluded that the increase in leaf thickness in *P. echinata* plants in full sun is attributed to the augmentation in their parenchymal and epidermal cell layers. In of *P. echinata* leaves, the palisade to spongy parenchyma ratio (PP:SP) was higher in the dry season, indicating enhanced light capture efficiency and utilization as previously reported in this species [45]. Leaf SP is a dynamic tissue that plays a vital role by helping maintain the water balance in situations of high transpiration rates, such as that undergone in high light and higher temperatures [46,47]. Since the mesophyll is the leading site of photosynthesis, adjustments on SP and the PP:SP ratio are associated with regulation of light absorption surface and CO_2_ diffusion surface.

The thicker cuticle observed in this study during the dry season in *P. echinata* is aligns with Gama et al. [45], who attributed this to increased radiance, which boosts cuticular transpiration and minimizes water loss even with closed stomata. Additionally, this thicker cuticle helps in light dissipation and heat management [7,48]. Generally, higher mesophyll thickness correlates with structural adaptations to sustain the photosynthetic process and improve water use efficiency due to increased net photosynthesis per leaf area [49]. Nonetheless, Gama et al. [45] describe that *P. echinata* plants in full sunlight developed lesser SLA with higher sclerophylly as an adaptation strategy. Baroni [15] describes that *P. echinata* have low morphological differences between plants that have been grown in 20% to 80% shade, which is confirmed by our studies.

Some scholars claim that the hypodermis might serve as a water storage tissue [7,39,50]; especially in arid climates, it is hypothesized that the hydrophilic nature of the cuticle is more critical for adaptation to xeric conditions than water storage in the hypodermis. In agreement with Rossatto and Kolb [51], dry-season leaves have a thicker cuticle and a higher proportion of wax than those that develop in a humid environment. Silva et al. [52] and Fernandes [39] emphasize that the cuticle, as well as the cutinized layers and the superficial wax, plays an essential role in reducing water loss.

Following González-Valenzuela et al. [53] and Zani et al. [54], the cuticle thickening is aligned with high irradiance or drought conditions as it increases cuticular resistance to transpiration, preventing water loss even when stomata are closed, and when present, it helps in the reflection of part of the light that falls on the leaves. In accordance with Gama et al. [45], we also noted that higher irradiance, typical of the dry season, is generally associated with lower relative humidity (Figure 1) and, consequently, higher stomatal density (Figure 5A), facilitating a water vapor layer around open stomata. This anatomical feature increases leaf resistance to water loss and reduces CO_2_ diffusion distance, optimizing carbon gain. The greater stomatal density (SD) with less-opened ostioles forms a water vapor layer around the leaf epidermis, enhancing resistance to water loss and CO_2_ diffusion to photosynthetic cells, thus maximizing net photosynthesis [54,55]. Carey [56] suggests that epidermal cell size in plants is influenced by both environmental and genetic factors, affecting the stomatal apparatus and index individually. In *C. nemorosa*, the ventral walls near the ostiole are thickened but smooth [39]. Stomata in *C. nemorosa* are larger (~476 μm^2^) compared to other Clusiaceae species studied by Fernandes [7], and they are classified as circular-like due to their shape ratio (1.28). Our study found stomatal density in *C. nemorosa* during the dry and rainy seasons to be 32% to 42% higher than that reported by Fernandes [39]. Additionally, a significant correlation exists between stomatal size and density, emphasizing their mutual relationship [39].

In plants under water deficit, maximum rates of electron transport and diurnal changes could suggest efficient decarboxylation of organic acids and refixation of carbon in the light [44]. Citrate breakdown as an internal CO_2_ source increases with drought [57,58]. There is some evidence that citrate may play different roles in different seasonal conditions. In light-limiting environments, it may function as a mechanism to save carbon and energy. However, in accordance with Borland et al. [59], drought stress did not affect citrate accumulation in young *Clusia aripoensis* leaves distinctly from findings for *P. echinata*, *C. glaziovii*, *L. tomentosa*, and to a lesser extent in *S. terebinthifolia*. Glucose and fructose levels declined significantly, indicating their predominant use for cellular respiration, confirmed by changes in sucrose levels. This metabolic shift highlights increased catabolic activity in response to prolonged water deficit, suggesting a concerted effort to repair photodamage encountered during resistance phases as previously reported by Mengarda et al. [11]. In *Clusia* species, soluble sugars and starch contribute to malate and citrate formation, similar to findings in accordance with Popp et al. [57] in *Clusia* sp. In terms of osmoregulation, raffinose, along with stachyose and verbascose, forms the Raffinose Family of Oligosaccharides (RFOs), synthesized from sucrose [60]. These compounds, along with mono- and disaccharides, act as antioxidants by scavenging reactive oxygen species (ROS) in plants, with synthesis increasing under water stress as potential osmoprotectors [10,11].

Comparative studies under constant light conditions (plasticity) may overestimate plant responses to radiation changes (acclimatization) by not accurately measuring biochemical effects or biomass distribution critical for acclimatization. Plasticity refers to phenotypic responses under varying environmental conditions, contrasting with constant conditions [61,62,63]. When comparing physiological features such as net photosynthesis and Fv/Fm of *P. echinata*, Baroni [15] describes high physiological plasticity in this species. High physiological (0.796) and metabolic (0.723) plasticity were also described in this study for *P. echinata*. Conversely, *S. terebinthifolius* exhibits low physiological plasticity in our study despite being considered highly plastic elsewhere [46] (Table 4).

Generally, two extreme groups of succession are distinguished: pioneer species, which grow only in clearings, and climax species, which require shaded understory environments to establish themselves [64]. However, there are many species occupying intermediate positions between these two classes. Thus, Budowski [65] and Cuzzol and Milanez [66] suggested the division of tropical trees into four successional groups: pioneers, early secondary, late secondary, and climax [64]. Studies suggest that tolerant species grow slower when compared to non-tolerant ones due to their lower metabolic rates [67,68]. From another perspective, the rapid growth in height when shaded can be considered an adaptation mechanism of competitive [68,69] plants. These plants are not able to tolerate low light intensities by readjusting their metabolic rates. Souza and Válio [70], in an experiment with fifteen tree species from the Atlantic Forest in different successional stages, observed a tendency for species from early stages of succession to show higher growth rates than late species, regardless of whether the environment received more or less light. Due to the scarce and contradictory results published about *P. echinata* and the different foliar forms found, a definitive classification of this species in terms of its functional group and ecological preferences for the intensity of irradiance of its foliar variants becomes complex [71].

## 4. Materials and Methods

### 4.1. Study Site and Plant Material

This study was conducted at the Caetés Ecological Station (ESEC-Caetés) in Paulista City, Pernambuco, Brazil (7°55′15″ to 7°56′30″ S and 34°55′15″ to 34°56′30″ W in accordance with Dos Santos et al. [72]). The ecological station is located in a lowland seasonal area of the Brazilian Atlantic Forest. The climate is classified as the “As” type in the Köppen–Geiger classification [19]. This climate type is characterized by short annual variations in atmospheric temperature and monthly precipitation ranging from 39.6 to 239.5 mm. According to historical precipitation data [73], August–February is the driest period, and March–July is the rainiest period. In the last 25 years, July was the rainiest month (388.41 ± 23.02 mm), while November was the driest (28.26 ± 4.16 mm). However, during the entire experimental period (June/2014–February/2015), 1059 mm of rainfall was registered (Figure 1). The relative humidity ranged from 69% to 76%. Mean monthly temperatures ranged from 20.5 °C in January to 26.4ºC in December. The global radiation intercepted by the plants varied widely during the experiment (Figure 1), reaching 14.9 MJ m^−2^ (June, the rainiest month) to 26.6 MJ m^−2^ (November, the driest month).

In agreement with Dos Santos et al. [72] chose five of the most abundant woody species (see Table 1). In addition, seedlings of all studied species, acquired from certified seedling producers, were planted according to their recommendations and randomly mixed in the study area. The spacing between plants was 2 × 2 m to standardize the experiment, summing 2500 plants per ha. The selection of the species was performed according to the following: (1) more than 15 individuals per species; (2) shoot height up to 150 cm; (3) the presence of at least one species from each successional group (i.e., pioneer and late successional); (4) economic and/or ecological potential of the species; and (5) no previous physiological, anatomical, and metabolic studies on the species. We additionally certified that all plants were at least 5 years old when the experiments were started.

### 4.2. Gas Exchange and Chlorophyll a Fluorescence

From June 2014 to February 2015, every two months, leaf gas exchange was determined on the 3rd attached fully expanded leaf from the apex, using a portable open-flow infrared gas analyzer (LI-6400XT; LI-COR Inc., Lincoln, NE, USA) with integrated fluorescence chamber heads (LI-6400-40; LI-COR Inc.). The net photosynthesis (*P*_N_, μmol CO_2_ g^−1^ s^−1^) and stomatal conductance (*g*_s_, mol H_2_O g^−1^ s^−1^) were measured on 5-year-old plants. All measurements were conducted at approximately 09:00–11:00 h solar time, under a clear sky and leaf irradiance of saturation of 1000 μmol m^−2^ s^−1^ (as previously tested by light curves versus *P*_N_), fixed CO_2_ concentration of 390 μmol mol^−1^, and airflow of 400 μmol s^−1^ [74]. The fluorescence analysis was carried out as described by Pompelli et al. [75].

### 4.3. Plant Anatomy

Leaf fragments of 5 cm^2^ of the 3rd attached fully expanded leaf from the apex (the same used in the previous topic) were sampled every two months. Two leaf fragments were collected from each plant, promptly immersed in FAA_50%_ for 48 h, and then stored in 70% (*v*/*v*) ethanol until analysis. Leaf samples were dehydrated in an ethylic series and embedded in plastic resin (Historesin-Leica Microsystems Nussloch, Heidelberg, Germany, part number 7592). Following this, all samples were processed as described in detail by Mendes et al. [76]. For each sample, 100 images were captured by a digital camera (Mikrosysteme Vertrieb GmbH, model ICC50 HD; Wetzlar, Germany) interfaced with a computer.

### 4.4. Metabolic Analysis and Metabolite Profile

Leaf samples of the 3rd leaflet were harvested in the same periodicity described above, flash-frozen in N_2,_ and stored at −80 °C until analysis. The samples were lyophilized before metabolite extraction. Approximately 25 mg of dry leaves was subjected to methanolic extraction as described by Silva et al. [77]. Sucrose, fructose, and glucose were measured according to the methodology described by Fernie et al. [78]. The methanolic insoluble fractions were analyzed for their total amino acids [44], and malate and fumarate were analyzed in accordance with Nunes-Nesi et al. [79]. The chlorophyll and total carotenoid contents were determined as previously described by Wellburn [80]. The levels of all other metabolites were quantified by gas chromatography–mass spectrometry (GC-MS) as previously described by Lisec et al. [81]. The mass spectra were cross-referenced with those in the Golm Metabolome Database [82]. The content of all compounds was normalized for dry mass and expressed in a heat map analysis and a putative metabolic pathway.

### 4.5. Plasticity Index

The plasticity index ranging from 0 (any plasticity) to 1 (full plasticity) was calculated as the difference between the average minimum and maximum for each sample divided by the maximum value [49] for physiological, anatomical, and metabolic features.

### 4.6. Experimental Design and Statistical Analyses

The experiments were conducted in a completely randomized block design with five different plant species, five sampled months, and five replicates. Two-way ANOVA was used to analyze all the data, and means were compared using an SNK test (*p* < 0.05) by Statistic version 14.0 (StatSoft, Tulsa, OK, USA). The correlations were made by Pearson correlations, using the value of all analyzed features. This dataset is presented as a Appendix A. Principal component analysis was performed after a multivariate analysis for all analyzed features in Minitab 18.1 (Minitab, Inc., Chicago, IL, USA). Heat maps were used to compare the means of each treatment, using the control (June, the rainiest month) as a reference. After the log2 transformation, the false color method, including a color scale, was used. The heat maps and a putative metabolic pathway were constructed using Microsoft^®^ Office 365 (https://www.microsoft.com/pt-br/microsoft-365, accessed on 5 July 2024) (Microsoft Corporation, Redmond, WA, USA) and Corel DRAW Graphics Suite X8 (Corel Corporation, Ottawa, ON, Canada).

## 5. Conclusions

In our study, we examined the potential of five tree native species, *Paubrasilia echinata*, *Chorisia glaziovii*, *Clusia nemorosa*, *Licania tomentosa*, and *Schinus terebinthifolia*, in a rainy gradient. To avoid high light intensities, such as those found in large gaps or degraded areas lacking vegetation cover, we recommend these species for enrichment planting in forest restoration projects. However, it is crucial to note that these recommendations are speculative. We must exercise caution when predicting the success of these species until maturity, as our investigation was limited to plants in their first years of development in an open full-sun area. Nonetheless, all the species studied show promise for enriching tropical primary (*C. glaziovii* and *C. nemorosa*) and secondary (*P. echinata*, *L. tomentosa*, and *S. terebinthifolia*) forests, provided they are planted in suitable light conditions according to their light tolerance. Our approach aims to identify key physiological traits that group species based on their light tolerance, thereby complementing existing knowledge on successional groups. This method can serve as a valuable guide for selecting the right species for enrichment plantings in tropical secondary forest restoration projects.

## Figures and Tables

**Figure 1 plants-13-01906-f001:**
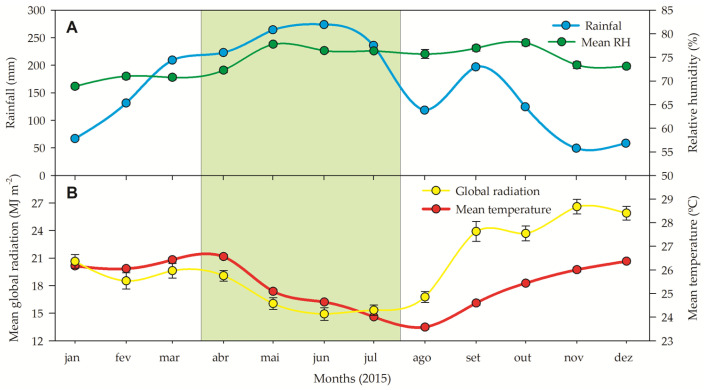
(**A**) Rainfall (blue symbols) and relative humidity (green symbols). (**B**) Global radiation (yellow symbols), and temperature (red symbols) measured during the experiment in a restructuring Brazilian Atlantic Forest fragment. All data were organized to show the rainy (green box) and dry season. Source: INMET [13], free access.

**Figure 2 plants-13-01906-f002:**
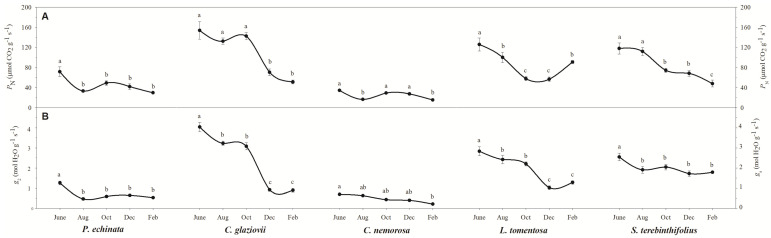
Net photosynthesis (*P*_N_; (**A**)) and stomatal conductance (g_s_; (**B**)) of *Paubrasilia echinata*, *Chorisia glaziovii*, *Clusia nemorosa*, *Licania tomentosa*, and *Schinus terebinthifolia* plants grown in the natural environment in 2014/June to 2015/Feb in accord of a historical rainfall gradient, in Paulista, Pernambuco, Brazil. Different letters denote significant differences for each month within each seasonal event (SNK, *p* ≤ 0.01). The values represent the mean of ten biological replicates (±SE).

**Figure 3 plants-13-01906-f003:**
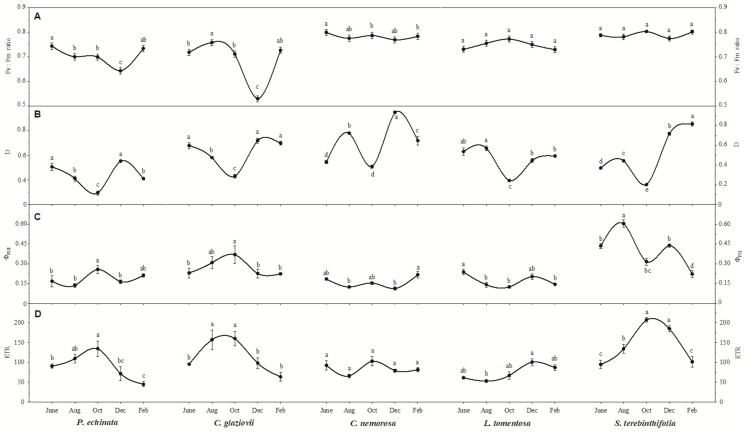
The ratio of variable-to-maximum chlorophyll fluorescence (Fv/Fm; (**A**)), fraction of absorbed photosynthetic active radiation dissipated as heat (**B**,**D**), actual PSII efficiency during the exposure to sunlight (Φ_PSII_; (**C**)), and electron transport rate (ETR; (**D**)) of *Paubrasilia echinata*, *Chorisia glaziovii*, *Clusia nemorosa*, *Licania tomentosa*, and *Schinus terebinthifolia* plants grown in natural environment in 2014/June to 2015/Feb in accord of a historical rainfall gradient, in Paulista, Pernambuco, Brazil. Different letters denote significant differences for each month within each seasonal event (SNK, *p* ≤ 0.01). The values represent the mean of ten biological replicates (±SE).

**Figure 4 plants-13-01906-f004:**
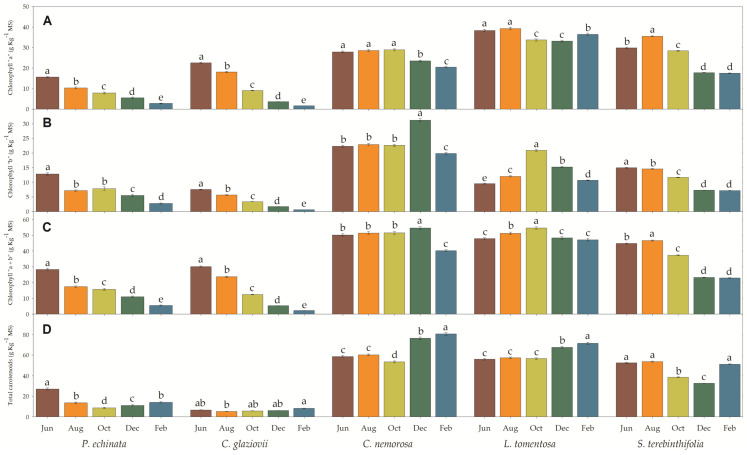
Chlorophyll “a” (**A**), “b” (**B**), total (**C**) and total carotenoids (**D**) of *Paubrasilia echinata*, *Chorisia glaziovii*, *Clusia nemorosa*, *Licania tomentosa*, and *Schinus terebinthifolia* plants grown in natural environment in 2014/June to 2015/Feb in accord of a historical rainfall gradient, in Paulista, Pernambuco, Brazil. Different letters denote significant differences for each month within each seasonal event (SNK, *p* ≤ 0.01). The values represent the mean (±SE) (n = 10).

**Figure 5 plants-13-01906-f005:**
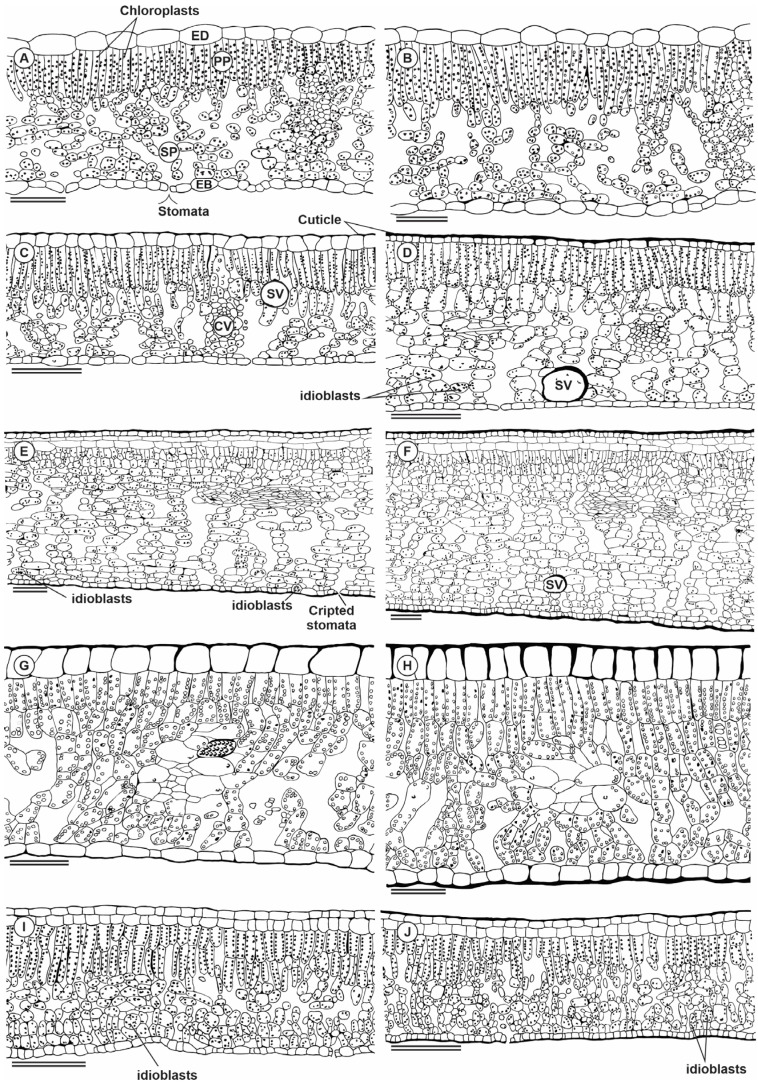
Illustration showing the cross-section of the *Paubrasilia echinata* (**A**,**B**), *Chorisia glaziovii* (**C**,**D**) *Clusia nemorosa* (**E**,**F**), *Licania tomentosa* (**G**,**H**), and *Schinus terebinthifolia* (**I**,**J**) plants grown in rainy (**A**,**C**,**E**,**G**,**I**) or dry (**B**,**D**,**F**,**H**,**J**) season in Paulista, Pernambuco, Brazil. ED, adaxial epidermis surface; EB, abaxial epidermis surface; PP, palisade parenchyma; SP, spongy parenchyma; SV, secretory vessel; CV, conduit vessel. Bars 100 μm.

**Figure 6 plants-13-01906-f006:**
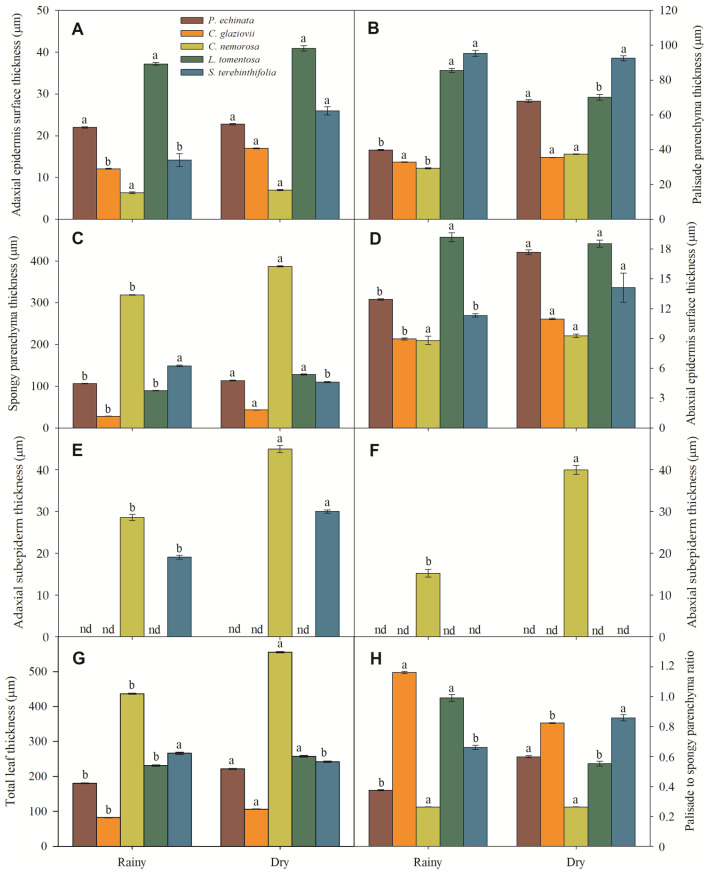
Adaxial epidermis surface thickness (**A**), palisade parenchyma thickness (**B**), spongy parenchyma thickness (**C**), abaxial epidermis surface thickness (**D**), adaxial subepidermis thickness (**E**), adaxial subepidermis thickness (**F**), total leaf thickness (**G**), and palisade to spongy parenchyma ratio (**H**) of *Paubrasilia echinata*, *Chorisia glaziovii*, *Clusia nemorosa*, *Licania tomentosa*, and *Schinus terebinthifolia* plants grown in natural environment in 2014/June to 2015/February in accord of a historical rainfall gradient, in Paulista, Pernambuco, Brazil. Different lowercase letters denote statistical differences between seasons within the same species (SNK, *p* ≤ 0.01). The values represent the mean (±SE) (n = 50).

**Figure 7 plants-13-01906-f007:**
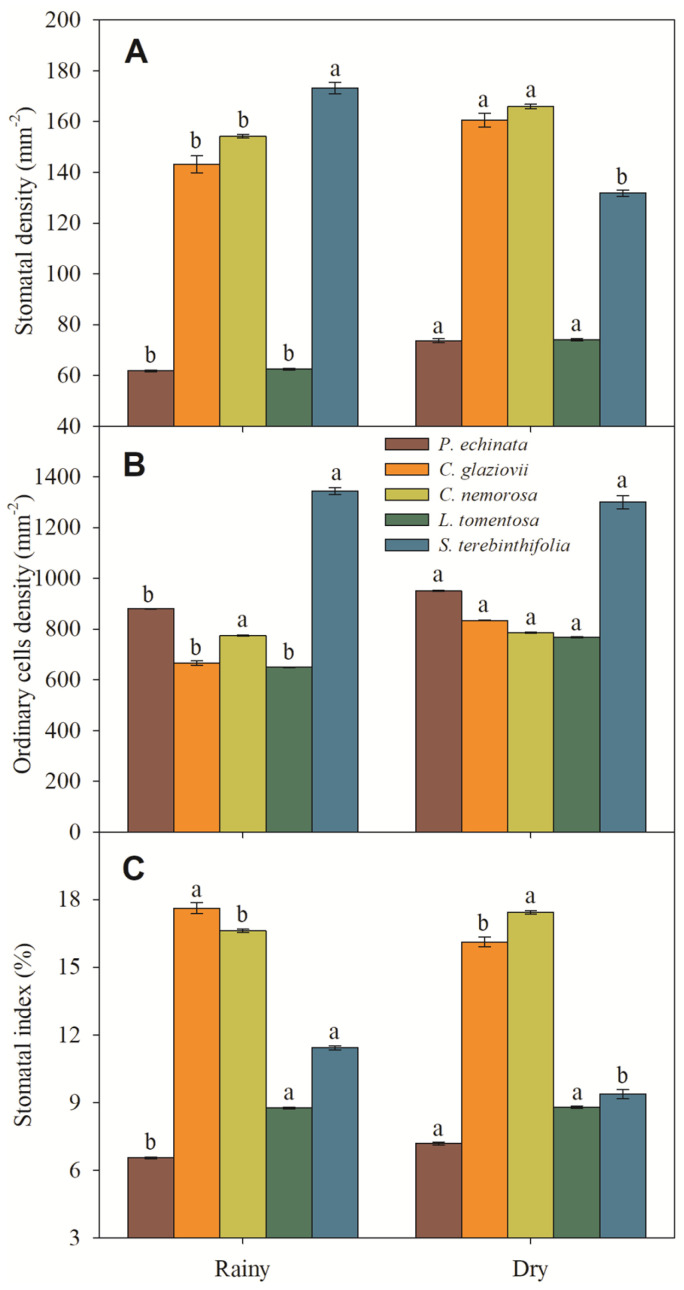
Stomatal density (**A**), ordinary cell density (**B**), and stomatal index (**C**) of *Paubrasilia echinata*, *Chorisia glaziovii*, *Clusia nemorosa*, *Licania tomentosa*, and *Schinus terebinthifolia* plants grown in natural environment in 2014/June to 2015/February in accord of a historical rainfall gradient, in Paulista, Pernambuco, Brazil. Different lowercase letters denote statistical differences between seasons within the same species (SNK, *p* ≤ 0.01). The values represent the mean (±SE) (n = 50).

**Figure 8 plants-13-01906-f008:**
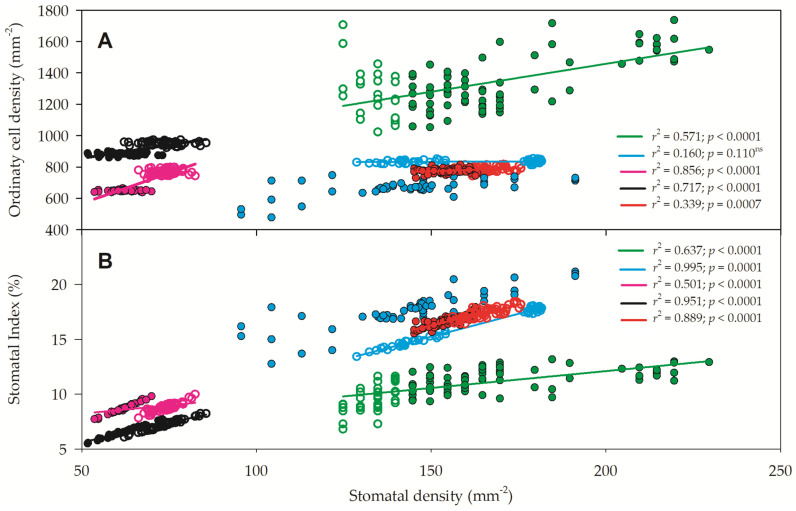
Relationship between stomatal density and ordinary cell density (**A**) and stomatal density and stomatal index (**B**) of *Paubrasilia echinata* (black symbols), *Chorisia glaziovii* (cyan symbols), *Clusia nemorosa* (red symbols), *Licania tomentosa* (pink symbols), and *Schinus terebinthifolia* (green symbols) plants grown in rainy (closed symbols) and dry (open symbols) natural environment in 2014/June to 2015/Feb in Paulista, Pernambuco, Brazil. Regression coefficients (*r*^2^) and *p* values are shown. (n = 50).

**Figure 9 plants-13-01906-f009:**
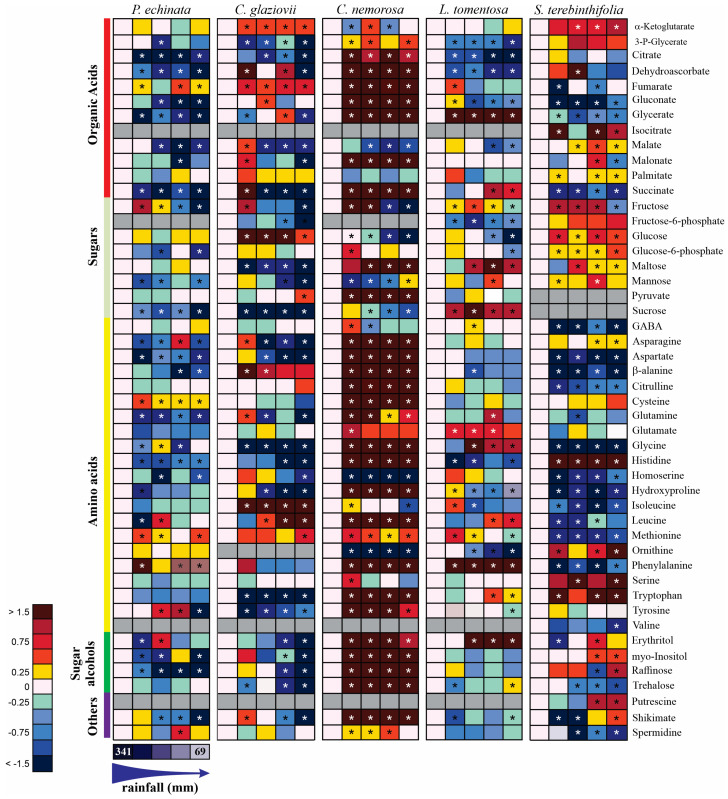
Heat map representing all metabolites in *Paubrasilia echinata*, *Chorisia glaziovii*, *Clusia nemorosa*, *Licania tomentosa*, and *Schinus terebinthifolia* plants grown in natural environment in 2014/June to 2015/February in accord of a historical rainfall gradient, in Paulista, Pernambuco, Brazil. The color code of the heat map is given at the log(2) scale. Data are normalized with respect to the mean response calculated for control rainfall (341, 114, 80, 72, and 69 mm in June/14, August/14, February/15, October/14, and December/14, respectively). To allow statistical assessment, individual plants from this set were normalized in the same way. Values represent means of five biological replicates. Cells in gray denote non-detectable metabolites. An asterisk (*) indicates that the values from other samples are significantly different from controls.

**Figure 10 plants-13-01906-f010:**
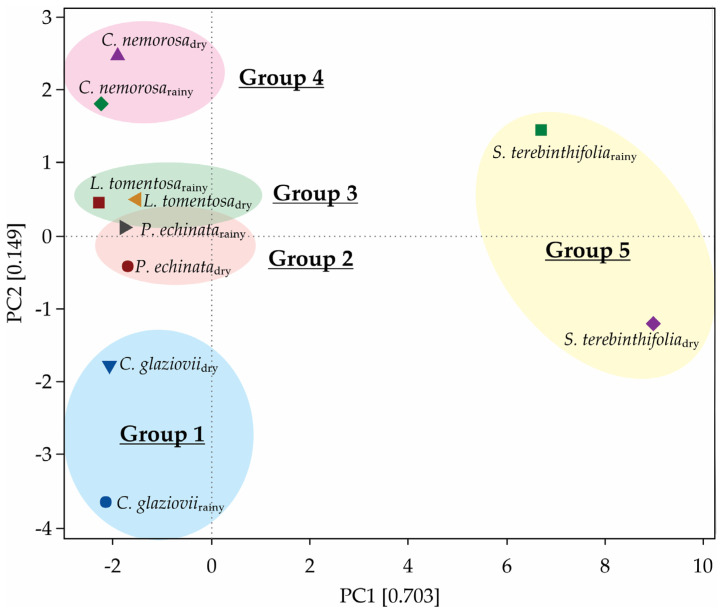
Principal component analysis showing the clustering of the species in accordance with similarities.

**Table 1 plants-13-01906-t001:** Photosynthetic metabolism, successional group, economic importance, and other information about all studied species.

Species	Photosynthetic Metabolism	Successional Group	Economic Importance	Tree Information	References
*Paubrasilia echinata Lam.—Gagnon*, *H.C.Lima & G.P.Lewis*	C_3_	Late Successional	Ornamental tree; its heartwood is used as dyestuff and to manufacture high-quality bows of stringed instruments, and the seeds have a pro-inflammatory activity. Used in silvicultural systems.	It is an endangered semideciduous tree species with a 10–15 m tall stalk 40–70 cm in diameter. Bipinnate leaves 10 to 15 cm in length and 5–6 pairs of pinnae. Some authors consider *P. echinata* as late successional [11,14]; however, Baroni [15] describes it as a pioneer.	Mengarda et al. [11], Borges et al. [16], Baroni [15]
*Chorisia glaziovii* (Kuntze) E. Santos)	C_3_	Pioneer	Extract from stem bark shows significant antimicrobial activity. Stem bark or leaves are used in folk medicine in the treatment of inflammation, diabetes, rheumatism.	A medium-sized tree 6–15 m tall, with composite leaves 3 cm in length.	Leal et al. [17], Albuquerque et al. [18]
*Clusia nemorosa* (G. Mey)	Many species facultative and constitutive metabolism CAM	Pioneer	It has pharmacological potential due to its antibacterial, antifungal, antiviral, anti-inflammatory, hepatoprotective, and antioxidant properties. Leaf extracts have anti-inflammatory and antiproliferative effects. Widely used in folk medicine.	*C. nemorosa* is found from the Brazilian Northern coast to Rio de Janeiro, and in forests and “Campo Rupestre” vegetation from the Amazon to Bahia. It is a 20 m tall tree; it has oblong and cartilage leaves. *Clusia* spp. are considered pioneers or facilitators for other species in the community.	Winter et al. [19], Trusheva et al. [20]
*Licania tomentosa* (Benth) Fritsch.	C_3_	Late Successional	This species has ethnopharmacological potential because it has numerous therapeutic attributes. Seed extracts of *L. tomentosa* have been demonstrated to exert inhibitory activity against the herpes simplex virus, and extracts of its leaves and fruits have been shown to have anti-cancer properties against leukemia cell strains.	It is a tree that has chemical features such as flavonoids, terpenoids (triterpenes and diterpenes), steroids, and tannins.	Rocha et al. [21], Fernandes et al. [22]
*Schinus terebinthifolia* Raddi	C_3_	Pioneer	Leaves, fruit, and bark have antimicrobial, analgesic, anti-inflammatory, antioxidant, anti-allergic, anti-free radical, and insecticidal activities. The fruits are rich in tannins, flavonoids, and essential oil and are used as cosmetology components, and in the perfume industry, presenting phenolic compounds and hydrolyzable tannins. It is used in protective reforestation.	It can be used in the recovery of degraded areas, as it demonstrates rapid growth and has a very expansive root system that facilitates water uptake from the deeper layers of the soil. The species has a remarkable anatomical plasticity. This characteristic is pivotal for its use in the early stages of ecological succession in land restoration. Also, it provides the species with an adaptive advantage in a climate change scenario. Its antioxidant enzyme activity is increased both in the aerial part and the root in response to water deficit.	Some authors consider this species as a pioneer, like Pilatti et al. [23] and Nunes et al. [24], while Dos Anjos et al. [25] consider it as early secondary

**Table 2 plants-13-01906-t002:** Summary of ANOVA for specific leaf area (SLA). Data were determined in five Atlantic Rainforest species and five different months. Source of variation (SV), degrees of freedom residuals (DF), mean squares (MS), F value (F), and *p* value (*p*).

Source of Variation	DF	SS	MS	F	*p*
Species	9	1.702	0.189	74.838	<0.001
Months	5	0.013	0.003	0.988	0.435
Residual	45	0.114	0.003		
Total	59	1.828	0.031		

**Table 3 plants-13-01906-t003:** Specific leaf area (SLA) determined in five Atlantic Rainforest species, regardless of months, as the ANOVA table shows in Table 2. Each value denotes a mean (±SE; n = 10). Values followed by different letters denote statistical differences at *p* ≤ 0.001.

Species	Mean (SE) (cm^2^ g^−1^)
*Paubrasilia echinata*	10.9 ± 0.5 b
*Chorisia glaziovii*	15.1 ± 0.3 a
*Clusia nemorosa*	5.1 ± 0.3 c
*Licania tomentosa*	9.9 ± 0.2 b
*Schinus terebinthifolius*	9.8 ± 0.4 b

**Table 4 plants-13-01906-t004:** Plasticity index of physiological, anatomical, and metabolic features in the leaves of *Paubrasilia echinata*, *Chorisia glaziovii*, *Clusia nemorosa*, *Licania tomentosa*, and *Schinus terebinthifolia* plants grown in natural environment in 2014/June to 2015/Feb in accord of a historical rainfall gradient, in Paulista, Pernambuco, Brazil. Means followed by lowercase letters denote statistical differences between species in the same feature (Newman–Keuls test *p* ≤ 0.001), and uppercase letters denote statistical differences between features (Bonferroni’s test at *p* ≤ 0.001). In metabolic features, the color denotes a distinct metabolic class: organic acids (red), sugars (light green), amino acids (yellow), sugar alcohols (dark green), and others (purple).

*Physiological Features*	*P. echinata*	*C. glaziovii*	*C. nemorosa*	*L. tomentosa*	*S. terebinthifolia*	Mean Value
Net photosynthesis (*P*_N_)	0.773	0.839	0.765	0.775	0.798	**0.790**
Stomatal conductance (*g*_s_)	0.837	0.877	0.879	0.815	0.638	**0.809**
Variable, maximum fluorescence (Fv/Fm)	0.233	0.406	0.145	0.181	0.122	**0.218**
Current PSII efficiency (Φ_PSII_)	0.981	0.820	0.795	0.814	0.836	**0.849**
Electron transport rate (ETR)	0.987	0.954	0.704	0.713	0.788	**0.829**
Active radiation dissipated as heat (D)	0.678	0.722	0.647	0.689	0.795	**0.706**
Chlorophyll “a”	0.894	0.938	0.392	0.259	0.539	**0.605**
Chlorophyll “b”	0.894	0.926	0.454	0.601	0.553	**0.686**
Chlorophyll “a + b”	0.855	0.935	0.366	0.243	0.539	**0.588**
Total carotenoids	0.830	0.454	0.430	0.314	0.436	**0.493**
**Mean Value**	**0.796 ± 0.069 a**	**0.787 ± 0.064 a**	**0.558 ± 0.074 b**	**0.540 ± 0.082 b**	**0.604 ± 0.069 ab**	**0.657 ± 0.061 B**
** *Anatomical features* **						
Abaxial epidermis surface thickness	0.462	0.429	0.857	0.880	0.948	**0.715**
Adaxial epidermis surface thickness	0.252	0.464	0.674	0.529	0.822	**0.548**
Palisade parenchyma thickness	0.531	0.345	0.289	0.888	0.707	**0.552**
Spongy parenchyma thickness	0.250	0.417	0.282	0.671	0.650	**0.454**
Total leaf thickness	0.305	0.284	0.262	0.370	0.427	**0.329**
Specific leaf area (SLA)	0.495	0.718	0.547	0.356	0.670	**0.557**
Ordinary cell density	0.116	0.440	0.106	0.201	0.394	**0.251**
Stomatal density	0.396	0.500	0.174	0.349	0.457	**0.375**
Stomatal index	0.332	0.397	0.158	0.226	0.483	**0.319**
**Mean Value**	**0.349 ± 0.045 a**	**0.444 ± 0.040 a**	**0.372 ± 0.087 a**	**0.497 ± 0.087 a**	**0.617 ± 0.064 a**	**0.456 ± 0.050 B**
** *Metabolic features* **						
α-Ketoglutarate	0.721	0.636	0.466	0.569	0.851	**0.649**
3-P-Glycerate	0.547	0.582	0.639	0.730	0.487	**0.597**
Citrate	0.958	0.972	0.767	0.841	0.869	**0.881**
Dehydroascorbate	0.801	0.910	0.994	0.678	0.730	**0.823**
Fumarate	0.504	0.864	1.000	0.673	0.577	**0.723**
Gluconate	0.774	0.975	0.987	0.745	0.850	**0.866**
Glycerate	0.844	0.961	0.970	0.951	0.988	**0.943**
Isocitrate	nd	nd	nd	nd	0.817	**0.817**
Malate	0.812	0.621	0.653	0.757	0.760	**0.721**
Malonate	0.802	0.684	1.000	nd	0.950	**0.859**
Palmitate	0.648	0.885	0.590	0.774	0.767	**0.733**
Succinate	0.908	0.997	0.992	0.780	0.960	**0.927**
Fructose	0.729	0.787	0.995	0.759	0.803	**0.815**
Fructose-6-phosphate	nd	0.609	nd	0.625	0.753	**0.662**
Glucose	0.510	0.990	0.386	0.809	0.692	**0.678**
Glucose-6-phosphate	0.809	0.909	0.612	1.000	0.538	**0.774**
Maltose	0.465	0.674	0.887	0.824	0.784	**0.727**
Mannose	0.621	0.899	0.686	0.699	0.960	**0.773**
Pyruvate	0.452	0.771	0.999	0.538	nd	**0.690**
Sucrose	0.680	0.798	0.693	0.788	nd	**0.740**
Aminobutyric acid (GABA)	0.603	0.646	0.578	0.543	0.992	**0.672**
Asparagine	0.862	0.935	0.999	0.595	0.554	**0.789**
Aspartate	0.834	0.966	0.996	0.999	0.992	**0.957**
β-alanine	0.824	0.825	0.995	0.618	0.554	**0.763**
Citrulline	0.501	0.575	1.000	0.713	1.000	**0.758**
Cysteine	0.768	0.758	0.999	0.709	0.950	**0.837**
Glutamine	0.770	0.929	0.729	0.824	0.745	**0.800**
Glutamate	0.814	0.454	0.626	0.707	0.731	**0.666**
Glycine	0.725	0.892	0.999	0.870	0.707	**0.839**
Histidine	0.649	0.992	0.895	0.838	0.839	**0.843**
Homoserine	0.901	0.877	0.904	0.685	0.990	**0.871**
Hydroxyproline	0.706	0.991	0.999	0.732	0.944	**0.874**
Isoleucine	0.541	1.000	0.673	0.761	0.986	**0.792**
Leucine	0.903	0.884	0.973	0.734	0.927	**0.884**
Methionine	0.621	0.652	0.538	0.604	0.817	**0.646**
Ornithine	0.569	nd	0.881	0.816	0.891	**0.789**
Phenylalanine	0.822	0.896	0.982	0.975	0.792	**0.893**
Serine	0.504	0.563	0.616	0.550	0.798	**0.606**
Tryptophan	0.588	0.993	0.802	0.608	0.853	**0.769**
Tyrosine	0.921	0.984	0.872	0.832	0.906	**0.903**
Valine	nd	nd	nd	nd	0.876	**0.876**
Erythritol	0.870	0.896	0.931	0.844	0.543	**0.817**
myo-Inositol	0.875	0.954	0.910	0.504	0.519	**0.752**
Raffinose	0.866	0.977	0.992	0.516	0.802	**0.831**
Trehalose	0.578	0.950	1.000	0.540	0.827	**0.779**
Putrescine	nd	nd	nd	nd	0.827	**0.827**
Shikimic acid	0.883	0.762	0.981	0.985	0.861	**0.894**
Spermidine	0.713	0.939	0.446	0.665	0.683	**0.689**
**Mean Value**	**0.723 ± 0.021 b**	**0.837 ± 0.022 a**	**0.833 ± 0.027 a**	**0.734 ± 0.025 b**	**0.812 ± 0.020 a**	**0.791 ± 0.014 A**

Nd, not detected.

## Data Availability

The data presented in this study are available upon request from the corresponding author.

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
