# Peer review of "Photosynthesis, Anatomy, and Metabolism as a Tool for Assessing Physiological Modulation in Five Native Species of the Brazilian Atlantic Forest"

_plants, 2024, doi:10.3390/plants13141906_

Round 1

Reviewer 1 Report

Comments and Suggestions for Authors

The concept of this research article is interesting and a major advantage of this research is the consideration of native woody plant species inhabiting the Brazilian Atlantic Forest with regard to their response to seasonal variations in rainfall. Generally, the manuscript is carefully prepared, the graphics are clear and the research methods are well described. Some minor specific comments on the content are listed below.

1)     Abstract: All tree species studied should be listed in the abstract.

2)     The sentences in lines 62-63 and 66-67 (page 2) are repeated.

3)     The figure 3 is a little bit unreadable, perhaps a change of resolution should be considered.

4)     The content of photosynthetic pigments is not calculated correctly. Chlorophyll a is generally 2 to 6 times more abundant than chlorophyll b, and carotenoids are even less abundant than chlorophyll b. These three pigment groups cannot be at similar levels. This should be corrected.

5)     For the mean values given in the text when describing the results, it is not necessary to give the standard error.

6)     To avoid distorting the differences between the averages, the y-axis of the figures should always start at "0".

7)     The discussion should be more concise.

8)     References should be limited to those that are essential and closely related to the topic of the manuscript - 109 references is far too many for a research article. Sometimes less is better.

Author Response

All questions and answers are presented in attachment 

Reviewer 2 Report

Comments and Suggestions for Authors

This paper tries to evaluate the strategies of several native species of Brazilian Atlantic Forest . It is a very extensive paper, with a lot of information and I think it is very interesting for the objective that they are proposing. It is a very exhaustive work with very interesting supplementary material.

 Regarding the formal aspects it is well written and presented. The material and methods are very extensive and the discussion is very wide-ranging. However, there are some questions that need to be dealt with.

On general aspects I would like to comment on several things. I think the discussion should be rewritten. There is a lot of data in the work of the authors and the discussion hardly comments on many of the results obtained; on the other hand, many works and references are mentioned that are not relevant.

Another aspect that is not clear to me is that the authors relate their results very much to work on plants growing in shaded and sunny conditions. I don't think that the kind of study they have carried out can be compared with those conditions. I don't think that shaded conditions can always be compared with rainy season conditions. You would have to be very careful to establish relationships here.

Figure 2: I think it makes no sense to analyse differences in photosynthesis and conductance between species. Each species has intrinsic values that are not comparable with other species. It is more interesting to compare the evolution of these parameters with respect to the seasons of the year and how the lack of rainfall affects each species.

Line 609-611: this affirmation is not correct. No species recorded higher photosynthetic rates in the dry season compared to the rainy season. The values for L. tomentosa in June are around 140 compared to 100 in February. Therefore this paragraph should be changed.

It should be noted that L. tomentosa recovers its photosynthetic rate in February, whereas the other species do not show this tendency.

Line 636-638: there is no figure relating the chlorophyll/carotenoid ratio. If so much importance is given to it in the text, perhaps it would be interesting to include this relation in Figure 4. Moreover, the variations of photosynthetic pigments are rarely mentioned. The results obtained should be discussed, where it can be seen how there are species that scarcely have differences throughout the year while others have a clear decrease in the dry season.

Line 658-661: this statement is not correct. If we look at figures 2, 3 and 4 there are very few differences in the ecophysiological parameters. The photosynthesis values are the same throughout the experiment and the conductance values are also the same except in February. There are also no differences in ETR, or Fv/Fm. The photosynthetic efficiency of PSII only varies in the month of February. And in many pigment data there are no differences either. In my opinion, most of the paragraph should be deleted. If the species C. nemorosa were CAM it would have no diurnal conductance. Moreover, as we have said, the data do not provide significant differences as the authors claim.

I would also like the authors to explain why the species C. glaziovii decreases its epidermis in the dry season, it seems to me a relevant data of figure 5. Also why are there species that increase their cuticle compared to others in which these changes are not seen. From what I observe while reading the article, much is said about the references of other authors, but very few of the results obtained by the authors are discussed. For example, in Figure 5 there is much discussion of CAM and C3 metabolism of species of the genus Clusia, 22 lines of text. And only C. nemorosa is mentioned. Also the following paragraph focuses exclusively on the genus Clusia. In contrast, there is absolutely no mention of the other species in the experiment.

Again, in the case of Figure 6, only the case of the species P. echinata is discussed.

Line 712: light does not diffuse. Change the sentence to regulation of light absorption surface and CO2 diffusion surface.

Indicate in the text what TLT means.

Line 174: in which part of this paper is the thickening of the cuticle of P. equinata discussed? Figure 5 does not show cuticle appearance in the dry season.

In summary, the paper has great potential and is very interesting, but I think it needs to be improved.

Author Response

(The authors gave the same response as above.)

Round 2

Reviewer 2 Report

Comments and Suggestions for Authors

After reading the authors' answers, I think the text has been sufficiently rewritten to correct the planned doubts.

Short comments:
Regarding the question raised by the authors, in the first version, lines 658-661 quoted: "Fernandes [7] describes non-significant differences between rainy and dry seasons leaf anatomy in C. criuva, data that diverges from those presented in this study to C. nemorosa where both physiological and anatomical differences were statistically different in rainy and dry season leaves." The authors commented that there were significant differences in physiological aspects in the case of C. nemorosa. If we look at figures 2 and 3, there were no significant differences.
In the new version, they have removed the reference to differences in physiological aspects from the text. This has been solved. Other solutions would have been to indicate that the authors did not detect significant differences in physiological aspects, and that these results were in agreement with Fernandes [39].

Author Response

Please see the letter to editor in attachment.

Best regards
